# Non-Invasive Differential Diagnosis of Cervical Neoplastic Lesions by the Lipid Profile Analysis of Cervical Scrapings

**DOI:** 10.3390/metabo12090883

**Published:** 2022-09-19

**Authors:** Alisa Tokareva, Vitaliy Chagovets, Djamilja Attoeva, Natalia Starodubtseva, Niso Nazarova, Kirill Gusakov, Eugenii Kukaev, Vladimir Frankevich, Gennady Sukhikh

**Affiliations:** 1National Medical Research Center for Obstetrics Gynecology and Perinatology Named after Academician V.I., Kulakov of the Ministry of Healthcare of Russian Federation, 117997 Moscow, Russia; 2Moscow Institute of Physics and Technology, 141700 Moscow, Russia; 3V.L. Talrose Institute for Energy Problems of Chemical Physics, Russia Academy of Sciences, 119991 Moscow, Russia; 4Translational Medicine Department, Siberian State Medical University, 634050 Tomsk, Russia

**Keywords:** mass spectrometry, cervical cancer, cervical dysplasia, diagnostics, lipids

## Abstract

Cervical cancer is one of the most common cancers in women with pronounced stages of precancerous lesions. Accurate differential diagnosis of such lesions is one of the primary challenges of medical specialists, which is vital to improving patient survival. The aim of this study was to develop and test an algorithm for the differential diagnosis of cervical lesions based on lipid levels in scrapings from the cervical epithelium and cervicovaginal canal. The lipid composition of the samples was analyzed by high-performance chromato-mass spectrometry. Lipid markers were selected using the Mann–Whitney test with a cutoff value of 0.05 and by projections to latent structures discriminant analysis, where a projection threshold of one was chosen. The final selection of variables for binomial logistic regressions was carried out using the Akaike information criterion. As a result, a final neoplasia classification method, based on 20 logistic regression sub-models, has an accuracy of 79% for discrimination NILM/cervicitis/LSIL/HSIL/cancer. The model has a sensitivity of 83% and a specificity of 88% for discrimination of several lesions (HSIL and cancer). This allows us to discuss the prospective viability of further validation of the developed non-invasive method of differential diagnosis.

## 1. Introduction

Cervical cancer (CC) ranks fourth among women’s cancers and second among women aged 15–44. According to a report by the International Agency for Research on Cancer (IARC), more than 604,000 new cases of cervical cancer were diagnosed worldwide in 2020, more than half of which were fatal [1].

The following tests are used to detect cervical neoplasia in clinical practice: cytological examination (liquid-based cytology (LBC), Pap test), HPV test, colposcopy, and biopsy, followed by histological examination. In some cases, analysis of p16 and Ki-67 markers by immunocytochemistry and immunohistochemistry is used to assess individual risk. The main purpose of a cytological examination is to identify precancerous diseases of the cervix for their timely treatment [2].

The correlation of the results of cytological and histological morphological studies is one of the most important criteria for the quality of CC screening. A study by Alanbay et al. showed that cytology results were inconsistent with subsequent histological diagnosis in 31% of cases [3]. R. Gupta et al. also analyzed the agreement between cytology and histology and found major discrepancies (incorrect differentiation absence of intraepithelial lesion and HSIL or cancer) in 6.4% of cases and minor discrepancies (one-step difference between cervical smear and cervical biopsy) in 20.4% of cases [4]. In the study of B.A. Crothers et al. discrepancies were noted in 20.4% of cases [5]. The multicenter KGOG 1040 study revealed 7.9% discrepancies between cytological classification into groups of negative for intraepithelial lesion or malignancy (NILM), atypical squamous cells of undetermined significance (ASCUS), low-grade squamous intraepithelial lesion (LSIL) and histological classification into groups of the high-grade squamous intraepithelial lesion (cervical intraepithelial neoplasia (CIN II–III) and cervical cancer) [6]. These results oblige researchers to search for and develop new, more accurate non-invasive clinical and diagnostic approaches to the diagnosis of neoplastic processes in the cervix. Today, there is much discussion about the involvement of new and rapidly developing omics studies such as metabolomics, proteomics, and transcriptomics [7]. Thus, significant efforts are being focused on developing a panel of genetic markers, which methylation may be associated with cervical neoplastic lesions [8,9]. Duraipandian et al. showed a difference in the molecular composition of tissues affected by precancerous lesions and normal tissues [10]. Paraskevaidi demonstrated LA-REIMS potential to differentiate CIN2+ grade lesions [11]. Furthermore, Porcaci et al. discovered metabolites that can accurately detect HSIL+ tissues [12].

Lipid metabolism disorder is one of the most prominent metabolic changes observed in cancer. Increased synthesis or uptake of lipids contributes to the rapid growth of cancer cells and the formation of tumors [13,14,15,16,17]. Thus, it is extremely important to study and evaluate the diagnostic potential of cervical lipidome changes in tissues with HPV-associated lesions, including CC, using mass spectrometry. Our group has previously created an accurate, rapid diagnostic method based on lipidome (ESI-MS analysis of a cervical epithelial tissue direct injection) [16]. This method can be further developed as an adjunct to histological analysis. However, cervical cancer screening requires a non-invasive approach, in particular, using samples of scrapings for cytological examination. The creation of a panel of lipid markers may allow accurate non-invasive differential diagnosis of the severity of cervical lesions and predict the outcomes of neoplastic transformation. The aim of this investigation was to study the possibility of non-invasive diagnosis of the cervix HPV-associated lesions based on the analysis of the lipid composition of the cervix epithelium.

## 2. Materials and Methods

The study included 111 patients aged 21 to 45 years (mean age 34 ± 9 years).

Inclusion Criteria:Age from 21 to 45 years;High-risk HPV;Low and high-grade squamous intraepithelial lesions, cervical cancer confirmed by histology examination;Regular menstrual cycle;Ability to comply with protocol requirements;Providing written informed consent to participate in the study.

Exclusion Criteria:Pregnancy;Lactation period;Hormone therapy;Acute inflammation;Dysfunction of the kidneys, liver, lungs in the stage of decompensation;Psychoneurological conditions.

We performed an examination of all patients, including cytological examination, cervical biopsy, histological examination of biopsy material, and lipidomic analysis of scrapings of the cervical epithelium. In cases of CC, cytological evaluation of smears was carried out according to the Bethesda system (2014).

Based on the results of biopsy material histological examination, 5 groups were formed:Group 1 (control)—NILM and HPV-positive (*n* = 8; 7.2%);Group 2—Chronical cervicitis and HPV-positive (*n* = 29; 26.1%);Group 3—LSIL (*n* = 32; 28.8%);Group 4—HSIL (*n* = 19; 17.1%);Group 5—Cervical cancer (*n* = 23; 20.7%).

Histological verification of the diagnosis was based on a two-level histopathological classification of CC precancerous processes. According to it, the term “mild epithelial dysplasia” (CIN I) corresponds to LSIL, and the term “moderate and severe cervical intraepithelial neoplasia” (CIN II–III) corresponds to high-grade squamous intraepithelial lesion (HSIL).

Scrapings from the cervical epithelium and cervical canal were taken with a cervical brush. The extraction of lipids from the collected biomaterial was carried out as follows: the brush was placed in an Eppendorf with 500 μL of water/methanol (1/1, *v*/*v*) and kept for 5 min in a Vortex, and then for 5 min in an ultrasonic bath. Then it was removed from the Eppendorf. After adding 1 mL of chloroform to the Eppendorf, it was kept in a Vortex for 10 min and then centrifuged for 5 min at 15,000 rpm. The lower organic layer with a volume of 950 µL was taken into a separate vial. After drying in a stream of nitrogen, the sample was redissolved in 200 μL of isopropanol/acetonitrile (1/1, *v*/*v*).

The lipid composition of the tissue was analyzed by LC/MS according to a previously developed protocol [18]. Samples were separated on a Dionex UltiMate 3000 chromatograph (Thermo Scientific, Waltham, MA, USA) coupled with Maxis Impact qTOF mass spectrometer (Bruker Daltonics, Bremen, Germany) with electrospray ionization. The analysis was performed in positive and negative ion modes [18]. Data preprocessing (Appendix A) and compound identification were carried out according to the protocol developed by J.P. Koelmel et al. by characteristic ion fragments (Appendix A) [19]. The lipid nomenclature corresponded to Lipid Maps [20]. The resulting LC/MS data were normalized using autoscaling (Appendix A) [21].

Statistical analysis was performed using scripts in the R language (4.1.1) [22] in the RStudio environment (1.383 GNU) [23] with packages “ropls” and “pROC”(Appendix A contains scripts).

The lipid features for logistic models for classification into any two of the clinical groups were selected in two ways. In the positive ion mode, at the first stage, lipids with *p*-value < 0.05 according to the Mann–Whitney statistical test were selected. In the negative ion mode, lipids with variable importance in projection (VIP) greater than 1 according to the orthogonal projection on latent structures discriminant analysis (OPLS-DA) (Appendix A contains statistical summary and S-plot, loading plot, and VIP-plot of OPLS-DA models) were selected. The second stage was the same for both positive and negative ion modes. The lipids selected in the first stage were used in logistic regression models and filtered by the Akaike information criterion [24]. The response variable was the group diagnosis. The milder case was assigned a value of “0” and the more severe case was assigned a value of “1”.

The final classification model was built based on 20 logistic regressions. There were 10 logistic regressions with lipids detected in positive ion mode for all pairwise combinations of the five studied diagnoses and 10 logistic regressions with lipids detected in negative ion mode. The final diagnosis was determined based on which of them received the most “votes”(Figure 1).

Each intermediate model was tested by cross-validation to determine its sensitivity, specificity, and threshold. The accuracy of each diagnosis was assessed as the ratio of the number of true cases of a diagnosis to the total number of cases with such a diagnosis given by the model.

## 3. Results

### 3.1. Lipidomic Profile

Histology is the gold standard for diagnosing cervical lesions. All other diagnostic methods are validated by histology. Table 1 demonstrates matches and differences between the cervix state classification results by cytology and histology for the samples studied. There were 29 patients with chronic cervicitis (further cervicitis) according to histological examination, and only 8 of them (27.6%) were revealed during a cytological examination. Similarly, the cytological examination revealed 18 patients with LSIL out of 32 patients with this diagnosis according to histology (56.3%), 10 patients with HSIL out of 19 (52.6%), and 18 patients with CC out of 23 (78.3%). Overall, cytological accuracy was 56%, sensitivity 76%, and specificity 88%.

The diagnostic potential of cervix lipid composition was studied. Lipid extracts from scrapings of the cervical epithelium were analyzed by liquid chromatography with mass spectrometry (HPLC/MS) in positive and negative ion modes. Characteristic chromatograms obtained for the studied groups are shown in Appendix A, and characteristic integrated mass spectra are shown in Appendix A.

In order to select lipids for diagnostic logistic regression models for discriminant analysis between two clinical groups, a pairwise comparison was performed using Mann–Whitney statistical test. In positive ion mode (Appendix A), 26 lipids were selected. Most lipids belonged to cholesterol esters, ceramides, lyso- and phosphatidylcholines, phosphatidylethanolamines, and triacylglycerols with an ether bond (Figure 2). Differences in the lipid composition of cervix epithelium when comparing the cervicitis control group with the LSIL and HSIL groups were expressed in a statistically significant decrease in the level of ceramides Cer-NDS d16:0/16:0 and Cer-NDS d16:0/18:0. In the CC group, compared to the control group, increased levels of phosphatidylcholine PC 16:0_18:1, PC 14:0_16:0, PC 16:0_20:4, PC 16:1_18:0, cholesterol esters CE 18:1, CE 24:1 and triacylglycerol TG 10:0_8:0_8:0 were observed. The analysis of cholesterol esters in cervicitis revealed a decrease in their levels in LSIL and HSIL for CE 18:1 and CE 24:1 and an increase in CE 18:1 in CC. The levels of phosphatidylcholines PC 16:0_22:6, PC 16:1_18:2 and sphingomyelins SM d18:0/16:0, SM d18:1/22:0 were also found to decrease in LSIL compared to cervicitis, while phosphatidylcholine PC 14:0_16:0, PC 16:0_16:0 and triacylglycerol TG 16:0_16:0_18:0, TG 14:0_16:0_16:1 levels increase in CC compared to cervicitis (Figure 2, Appendix A).

In negative ion mode (Appendix A), 36 lipids were selected for inclusion in diagnostic logistic regression models. The selected lipids belonged to lyso- and phosphatidylcholines, sphingomyelins, and oxidized phospholipids (Figure 3). There was a statistically significant increase in the levels of phosphatidylcholines PC O-16:0/16:0, PC 16:0_18:2, and sphingomyelin SM d16:0/18:2 in CC compared to those in the NILM group. The cervicitis and LSIL groups had higher levels of lysophosphatidylcholine LPC 18:2 compared to the control group (NILM). There is a significant increase in the levels of phosphatidylcholines and sphingomyelins SM d22:0/20:3, SM d24:1/18:1, SM d20:0/14:0 in CC compared to cervicitis (Figure 3, Appendix A).

With malignant transformation of the epithelium of the cervix, the level of particular phosphatidylcholines and sphingomyelins increased (PC 16:0_22:6, PC 14:0_16:0, SM d18:1/20:0 in HSIL vs LSIL groups, and PC 16:0_18:1, LPC 22:3, PC 16:0_20:5, SM d20:0/22:0 in CC vs LSIL). Lipid markers that differentiate precancerous HSIL and cervical cancer (CC) included lipid esters LPC O-16:0, PE P-16:0/20:4, and sphingomyelins SM d16:0/18:1, SM d22:0/20:3, SM d18:0/16:0, levels of which increased in cancer (Figure 2 and Figure 3, Appendix A).

Based on the above lipids, a set of discriminative models was developed. Detailed parameters of the models are given in Table 2 and Appendix A, including the area under the curve (AUC) with its confidence interval (CI), the threshold of the model, and its sensitivity and specificity. The discriminative models were used to build a classification algorithm, the accuracy of which was 79% (Table 3). The method demonstrated high accuracy in differentiating cervicitis and CC from NILM. The diagnosis of severe lesions (HSIL+) had a sensitivity of 83% and specificity of 88%. The lipidome-based model can be used to refine the diagnosis, and the agreement was higher for histology results than cytological diagnosis (Table 4).

### 3.2. Clinical Cases

A set of clinical cases were analyzed in more detail to demonstrate the application of the developed models (Figure 1). We re-classified cervical lesions into two groups: NILM, cervicitis, and LSIL—low lesion; HSIL and cancer—high lesion. The probability of belonging to each new class was calculated as division votes for the class (for example, votes for the high lesion is the sum of votes for HSIL and cancer) to sum all votes. The results are presented in Table 5.

#### 3.2.1. Clinical Case No. 1

Patient II (Table 4) had an HSIL cytology result, and HPV type 51 was identified. Colposcopy results revealed mild abnormalities in the epithelium of the cervix (Appendix A). The epithelial scraping lipidome analysis indicated the absence of neoplasia (Table 5). A biopsy of the cervix was performed. The histological diagnosis resembled chronic cervicitis. This clinical case demonstrates a significant discrepancy between cytology and histology results, with good agreement between histology and lipid model. The latter gives an 85% probability of low-grade lesion (cervicitis or LSIL) and a 15% probability of high-grade lesion (HSIL or CC) (Table 5).

#### 3.2.2. Clinical case No. 2

The patient designated VIII in Table 4 had an LSIL cytology result. HPV type 16 was detected. Colposcopy results revealed mild abnormalities in the epithelium of the cervix (Appendix A). The epithelial scraping lipidome analysis revealed HSIL (Table 5). A biopsy of the cervix was performed. The histological diagnosis resembled HSIL (CIN II-III). The treatment was carried out. According to Table 5, the probability of high lesions (HSIL and CC) in this patient was also higher than low ones—60% and 40%, respectively, which is consistent with the histological diagnosis.

## 4. Discussion

It is known that lipid metabolism disorders are closely associated with carcinogenesis. Cholesterol esters and cholesterol accumulate in macrophages [25], which engulf dead cells. Previously, based on studies of kidney cancer, it was shown that cholesterol metabolism is disturbed in cancer cells [26]. Neutral sphingomyelinase decomposes sphingomyelins into ceramides and choline phosphate. It is involved in the activation of signals associated with inflammatory processes and acts as a tumor suppressor [27,28]. Phosphatidylcholines are associated with the carcinogenesis process [29,30]. Similarly, A.M. Porcari et al. demonstrated differences in levels of ceramides and sphingosine metabolites in normal cervical tissues and tissues with severe neoplasia, as well as the potential utility of these markers for cervical tissue differentiation [12].

Our final model has a higher specificity for CC and high predictive value for predicting cervicitis and cervical cancer, as well as an average value for predicting LSIL relative to the NILM, compared to models based on p16/Ki-67 proteins immunocytochemistry, microRNA, and HPV DNA [31]. It should be noted that the above-mentioned methods are based on compounds associated with HPV and neoplastic transformations, which makes it difficult to differentiate lesions with low and no viral load (LSIL/cervicitis/NILM). The model, based on information obtained by Raman spectroscopy, demonstrates high accuracy in the differentiation of no lesions/CIN1/CIN 2-3 lesions/CC; however, this method is more invasive since it requires tissue biopsy [32,33]. Our previous study demonstrated 88% sensitivity and 71% specificity of the OPLS-DA model used to classify mild neoplastic lesions of the cervix (LSIL) from severe precancerous and malignant diseases of the cervix (HSIL and SCC) [16]. This model was based on the lipid profile of cervical tissues (biopsy samples). The non-invasive approach proposed in this study has an even higher diagnostic potential (83% sensitivity and 88% specificity).

Models based on lipid markers in serum also have high diagnostic potential for differentiating NILM and SIL [34]. Nam M et al. conducted a comparative study of plasma lipidome in HSIL, CC, LSIL, and control groups by mass spectrometry and found a difference in plasma lipid profile for each of the four groups [35]. Khan I, Nam M, and Kwon M found a difference in the polar metabolite levels in blood plasma in cases with no neoplastic cervical lesions and CIN1, CIN 2/3, CC [36].

## 5. Conclusions

The developed model, based on the lipid profile obtained by HPLC-MS analysis of cervical epithelium scrapings, makes it possible to differentiate HSIL and CC from LSIL, cervicitis, and NILM. The lipidome diagnostic panel in patients with HSIL and cervical cancer is characterized by the predominance of lipid classes of phosphatidylcholines and sphingomyelin, as well as cholesterol esters. The non-invasive method used in the work demonstrated a high diagnostic value for severe lesions (HSIL+) with 83% sensitivity and 88% specificity. The diagnostic potential for the lipid profile models was higher than cytological examination, shown in two clinical cases. This suggests the potential applicability of the developed method in cervical cancer screening.

## Figures and Tables

**Figure 1 metabolites-12-00883-f001:**
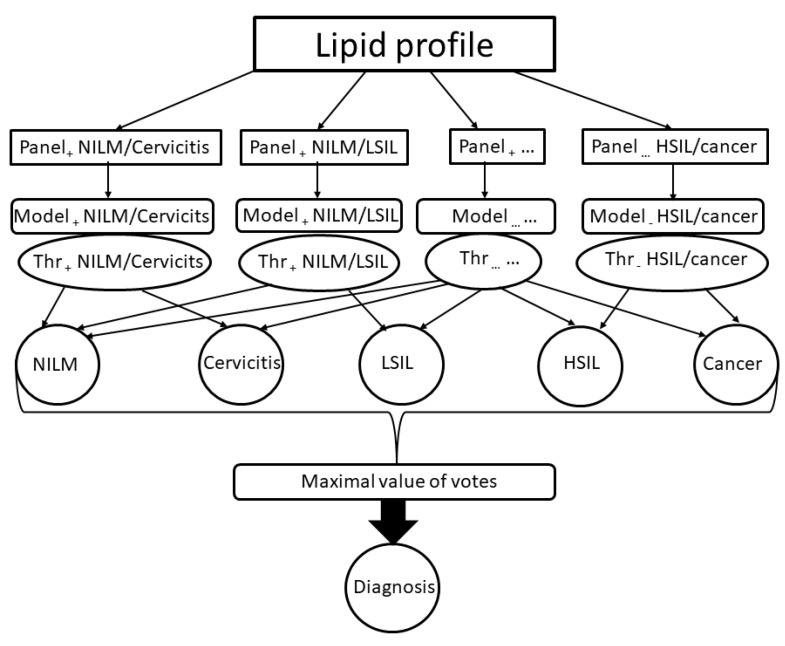
The structure of the final diagnostic model. Panel_x_ D1/D2 indicate lipids included in the logistic regression to discriminate diagnosis D1 from diagnosis D2 in the x ion mode; Model_x_ D1/D2 indicate model based on logistic regression to discriminate diagnoses D1 and D2 in the x ion mode, Thr_x_ D1/D2 indicate a threshold of a logistic regression result to set a D1 or D2 diagnosis.

**Figure 2 metabolites-12-00883-f002:**
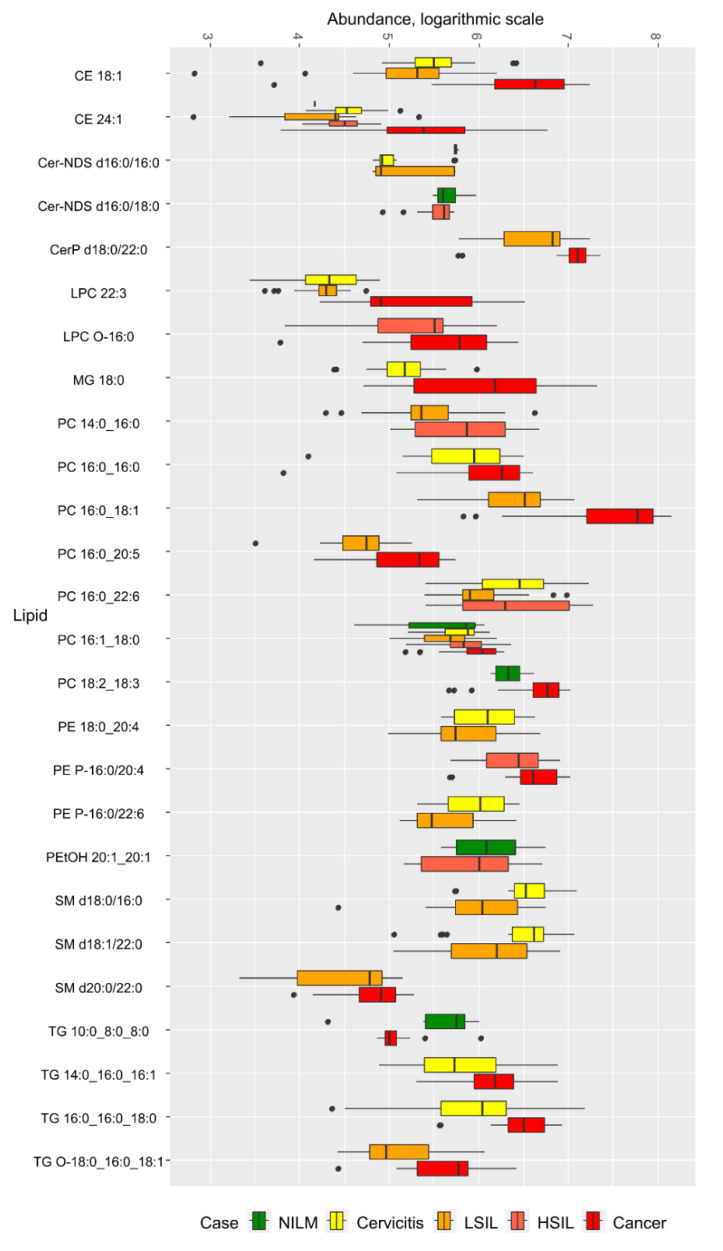
Box plot of lipid levels detected in positive ion mode and included in the discriminative models. CE—cholesterol ester; Cer-NDS—no-hydroxy fatty acid dihydrosphingosineceramide; CerP—ceramide-1-phosphate; LPC—lysophosphatidylcholine; LPC O—plasmanyllysophosphatidylcholine; MG—monoacylglycerol; PC—phosphatidylcholine; PE—phosphatidylethanolamine, PE P—plasmenylphosphatidylethanolamine; PEtOH—phosphatidylethanol; SM—sphingomyelin; TG—triacylglycerol; TG O—plasmanyltriacylglycerol.

**Figure 3 metabolites-12-00883-f003:**
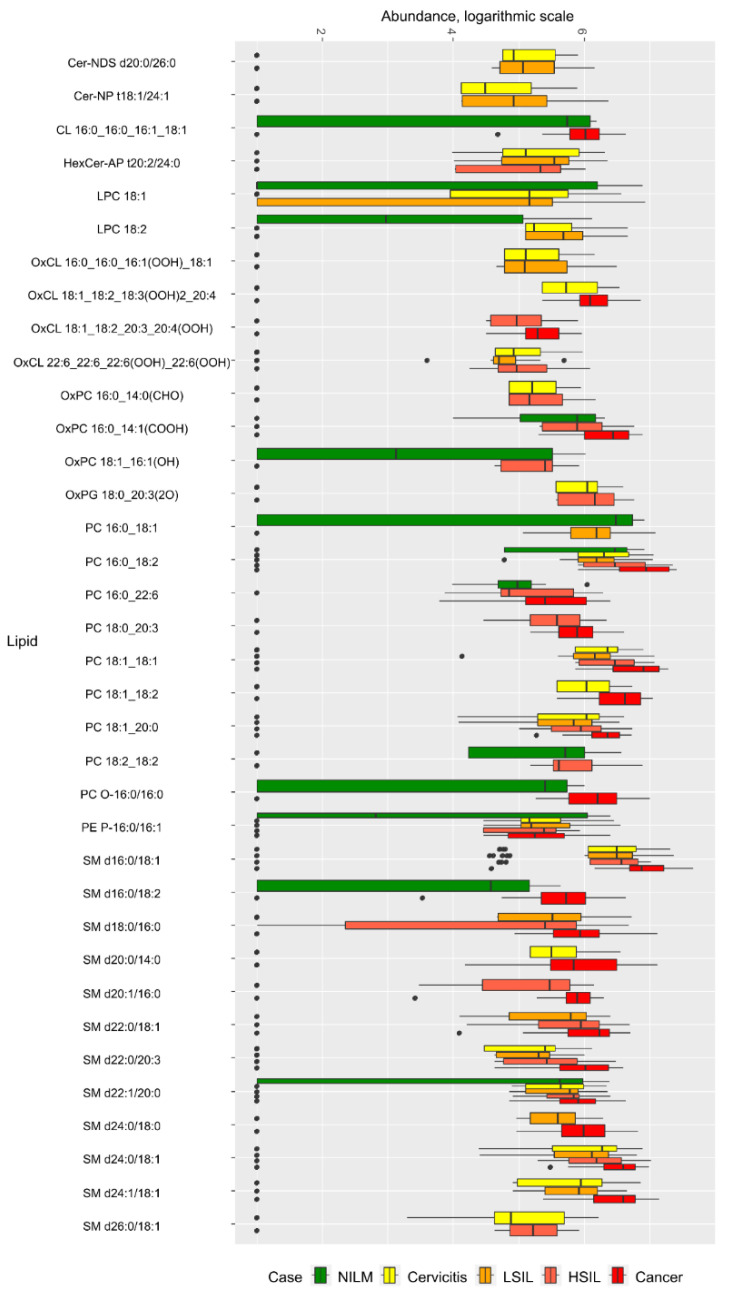
Box plot of lipid levels included in the discriminative models in the negative ion mode. Cer-NDS—non-hydroxy fatty acid dihydrosphingosineceramide; Cer-NP—non-hydroxy fatty acid phytosphingosineceramide; CL—cardiolipin; HexCer-AP—alpha-hydoxy fatty acid phytosphingosineglucosylceramide; LPC—lysophosphatidylcholine; OxCL—oxidized cardiolipin; OxPC—oxidized phosphatidylcholine; PC—phosphatidylcholine; PC O—plasmanylphosphatidylcholine; PE P—plasmenylphosphatidylethanolamine; SM—sphingomyelin.

**Table 1 metabolites-12-00883-t001:** Contingency table for diagnoses of cervix lesions established by cytology and histology. Number of patients and respective accuracy (%). The accuracy of each diagnosis is indicated in parentheses, calculated as the ratio of the number of true cases of any cytological diagnosis to the total number of cases with such a cytological diagnosis.

Histological Diagnosis	Cytological Diagnosis				
	NILM	Cervicitis	LSIL	HSIL	Cancer
NILM	8 (29)	0	0	0	0
Cervicitis	12	8 (53)	7	2	0
LSIL	3	5	18 (64)	6	0
HSIL	4	2	3	10 (45)	0
Cancer	1	0	0	4	18 (100)

**Table 2 metabolites-12-00883-t002:** Parameters of discriminative models based on logistic regression obtained during cross-validation.

Ion Mode	Model	AUC (CI AUC)	Thr. ^1^	Sens. ^2^, %	Spec. ^3^, %
Positive mode	NILM/cervicitis	0.89 (0.79–1.00)	0.97	100	67
	NILM/LSIL	0.93 (0.85–1.00)	0.96	100	57
	NILM/HSIL	0.68 (0.46–0.89)	0.81	100	44
	NILM/cancer	0.94 (0.87–1.00)	0.99	100	80
	Cervicitis/LSIL	0.81 (0.70–0.93)	0.13	74	95
	Cervicitis/HSIL	0.63 (0.46–0.81)	0.46	61	73
	Cervicitis/Cancer	0.90 (0.76–1.00)	0.99	00	85
	LSIL/HSIL	0.68 (0.52–0.84)	0.37	67	75
	LSIL/Cancer	0.89 (0.71–0.98)	0.01	91	88
	HSIL/Cancer	0.95 (0.85–1.00)	0.99	100	86
Negative mode	NILM/cervicitis	0.81 (0.56–1.00)	0.49	93	86
	NILM/LSIL	0.78 (0.56–1.00)	0.50	91	71
	NILM/HSIL	0.81 (0.66–1.00)	0.48	89	75
	NILM/cancer	0.75 (0.40–1.00)	0.99	87	63
	Cervicitis/LSIL	0.65 (0.51–0.79)	0.51	67	63
	Cervicitis/HSIL	0.82 (0.67–0.95)	0.15	64	87
	Cervicitis/Cancer	0.94 (0.86–1.00)	0.01	92	96
	LSIL/HSIL	0.64 (0.48–0.81)	0.39	56	73
	LSIL/Cancer	0.89 (0.73–0.99)	0.99	95	89
	HSIL/Cancer	0.77 (0.63–0.92)	0.27	76	76

^1^ Thr.—threshold; ^2^ Sens.—sensitivity; ^3^ Spec.—specificity.

**Table 3 metabolites-12-00883-t003:** Lipidome diagnostic contingency table for differential diagnosis of the severity of neoplastic lesions. Values in parentheses indicate the accuracy of each diagnosis.

	NILM	Cervicitis	LSIL	HSIL	Cancer
NILM	8 (0.89)	0	0	0	0
Cervicitis	0	23 (0.82)	2	4	0
LSIL	0	5	23 (0.74)	4	0
HSIL	1	0	6	12 (0.57)	0
Cancer	0	0	0	1	22 (1.00)

**Table 4 metabolites-12-00883-t004:** Comparison of lipidome, cytological and histological analysis of several patients. Bold type indicates patients for whom histology and lipidomic diagnosis coincided.

Patients	Histology	Cytology	Lipidome
I	HSIL	Cervicitis	LSIL
**II**	**Cervicitis**	**HSIL**	**Cervicitis**
III	HSIL	NILM	LSIL
**IV**	**HSIL**	**LSIL**	**HSIL**
V	HSIL	NILM	Cervicitis
VI	HSIL	NILM	LSIL
VII	HSIL	LSIL	Cervicitis
**VIII**	**HSIL**	**LSIL**	**HSIL**

**Table 5 metabolites-12-00883-t005:** Distribution of votes for each diagnosis in clinical cases for patients II and VIII.

Patient	NILM	Cervicitis	LSIL	HSIL	Cancer	Probability of Low Lesions, %	Probability of High Lesions, %
II	3	8	6	2	1	85	15
VIII	4	3	1	7	5	40	60

## Data Availability

Data are contained within the Appendix A.

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
