# Peer review of "Non-Invasive Differential Diagnosis of Cervical Neoplastic Lesions by the Lipid Profile Analysis of Cervical Scrapings"

_metabolites, 2022, doi:10.3390/metabo12090883_

Round 1

Reviewer 1 Report

The manuscript established an algorithm for the differential diagnosis of cervical lesions based on lipid levels in scrapings from the cervical epithelium and cervicovaginal canal. And this approach has been validated with an accuracy of 79%, a sensitivity 83% and a specificity 88%. The result is quite straight forward and solid. And the manuscript provided the possibility of a potential non-invasive method of differential diagnosis of cervical lesions.

1. Are there any unknown peaks (not identified) presented in the samples? If so, have any of the peaks been considered significant in the differentiate models?

2. Figures for the models, e.g., VIP score, S-plots, X2Y plots, should be provided in either the manuscript or the supplementary materials.

Author Response

The manuscript established an algorithm for the differential diagnosis of cervical lesions based on lipid levels in scrapings from the cervical epithelium and cervicovaginal canal. And this approach has been validated with an accuracy of 79%, a sensitivity 83% and a specificity 88%. The result is quite straight forward and solid. And the manuscript provided the possibility of a potential non-invasive method of differential diagnosis of cervical lesions.

  1. Are there any unknown peaks (not identified) presented in the samples? If so, have any of the peaks been considered significant in the differentiate models?

Answer: Yes, there are 222 detected peaks were not identified in positive ion mode and 1977 - in negative ion mode. We didn’t test these peaks for significance for diagnostics. In this study, we focused on the changes in the lipid profile of scrapings of the cervical epithelium due to neoplastic transformation. At the first stage, lipid identification was carried out, at the second stage, models were built based on the levels of identified lipids. Indeed, peaks not identified as lipids could contribute to the models, but this analysis is planned to be carried out in a subsequent detailed study of the features of the molecular composition of the transformed cervical epithelium (water-methanol extracts of scrapings were also saved for this).

  1. Figures for the models, e.g., VIP score, S-plots, X2Y plots, should be provided in either the manuscript or the supplementary materials.

Answer: Figures were added in supplementary materials as Supplementary file 2.

Reviewer 2 Report

The paper deals with identification/quantification of lipidome of cervical lesions. The work is based on the analysis of 29 patient samples and the resulting data is used to build models using OPLS-DA. 

The description of the method and the results is good, with only an error on line 68 ("tissue y direct"). Please correct. 

The authors previously worked with similar samples and also published OPLS-DA based statistical analysis, still it would be welcome, if the details of the method would appear in this paper as well. There is no mention about model validation either, which is critcal when working with low sample numbers. ropls library contains such functions, but also PCA is a good indicator of model suitability accortding to the 2016 paper PCA as a practical indicator of OPLS-DA model reliability,  by Bradley  Worley and Robert Powers.

Please comment on this / show model validation. 

My last comment is related to Table 5, where probability of classification is mentioned. Please describe how these numbers were calculated and why there is no probability of the samples falling into other classes. 

Author Response

The paper deals with identification/quantification of lipidome of cervical lesions. The work is based on the analysis of 29 patient samples and the resulting data is used to build models using OPLS-DA. 

The description of the method and the results is good, with only an error on line 68 ("tissue y direct"). Please correct. 

Answer: Corrected (line 70)

The authors previously worked with similar samples and also published OPLS-DA based statistical analysis, still it would be welcome, if the details of the method would appear in this paper as well. There is no mention about model validation either, which is critcal when working with low sample numbers. ropls library contains such functions, but also PCA is a good indicator of model suitability accortding to the 2016 paper PCA as a practical indicator of OPLS-DA model reliability, by Bradley  Worley and Robert Powers.

Please comment on this / show model validation. 

Answer: Statistical characteristics and figures of the OPLS model were added in supplementary file (Supplementary file 2). We calculate sensitivity, specificity and threshold of each model by leave-one-out cross validation. Accuracy of final model was calculated by leave-one-out-cross-validation. This information was added in lines 151-154: “Each intermediate model was tested by cross-validation to determine its sensitivity, specificity and threshold. The accuracy of each diagnosis was assessed as the ratio of the number of true cases of a diagnosis to the total number of cases with such a diagnosis given by the model.”

My last comment is related to Table 5, where probability of classification is mentioned. Please describe how these numbers were calculated and why there is no probability of the samples falling into other classes. 

Answer: Comments about probability of classification, describing of class and calculation methods were added in lines 244-248: “We re-classified cervical lesions in two groups: NILM, cervicitis and LSIL – low lesion; HSIL and cancer – high lesion. Probability of belonging to each new class was calculates as division votes for the class (for example, votes for high lesion is sum of votes for HSIL and cancer) to sum all votes.”.

Reviewer 3 Report

Dear Authors, I recommend the manuscript for publication, but before, please verify the few comments I made in the PDF file. Maybe, it can be interesting for you and to improve the manuscript overall aspect.

For example:

Line 45: I found some vague definitions, please define which kind of discrepancies, then we can understand the major and the minor ones.

Line 75: The short paragraph there is unusual. What do you think about to conect with the above paragraph, or improve the paragraphy by adding more description of this study, hypotesis and so on....

Line 79: Is unecessary comment "by far", if it is a gold standard for diagnosis. Then the following sentence can be removed. And, please rewrite the setence "Table 1 compares of the classification of cervix state by cytology and histology for the studied samples" - it is confusing "compares of the classification"

Table 1: Add the caption information: "Number of patients and respective accuracy (%)"

To conect the "previous" results with Lipids analys the phrase "The diagnosis of cervix lesions based on its lipid composition was considered in the present study" - should be rewriten

Line 108: Suggestion: this description can be improved, Avoid introduce to many values in text in sequence - It is already shown in the table, and comment the general behaviour - keep the table as reference

The caption of Figure 1 and 2 could be improved by describing TG, SM, and so on....

Table 2 - please provide the values in %

2.2. I think you can rewrite, that paragraph is not necessary as it is written.

Most of the readers probably don´t know R or want to use so Could you provide the code as support info? Or did you use a specific library? can you make a more detailed description about that? - It is also important to people understand the data processing and verify/reproduce the study/application.

Finally, I think you should reproduce this study by using Raman or FTIR spectroscopy to analyse the samples and built a prediction model with PCA+ML.

Author Response

Dear Authors, I recommend the manuscript for publication, but before, please verify the few comments I made in the PDF file. Maybe, it can be interesting for you and to improve the manuscript overall aspect.

For example:

Line 45: I found some vague definitions, please define which kind of discrepancies, then we can understand the major and the minor ones.

Answer: Definition of major (incorrect differentiation absence of intraepithelial lesion and HSIL or cancer) and minor (one-step difference between cervical smear and cervical biopsy) discrepancies was added (lines 45-47): “R. Gupta et al. also analyzed the agreement between cytology and histology and found major discrepancies (incorrect differentiation absence of intraepithelial lesion and HSIL or cancer) in 6.4% of cases and minor discrepancies (one-step difference between cervical smear and cervical biopsy) in 20.4% of cases”.

Line 75: The short paragraph there is unusual. What do you think about to conect with the above paragraph, or improve the paragraphy by adding more description of this study, hypotesis and so on....

Answer: The paragraph with aim were connected with above paragraph.

Line 79: Is unecessary comment "by far", if it is a gold standard for diagnosis. Then the following sentence can be removed. And, please rewrite the setence "Table 1 compares of the classification of cervix state by cytology and histology for the studied samples" - it is confusing "compares of the classification"

Answer: “by far” was excluded (line 157). The sentence "Table 1 compares of the classification of cervix state by cytology and histology for the studied samples" was changed (line 158-159): “Table 1 demonstrates matches and differences between the cervix state classification re-sults by cytology and histology for the samples studied.”.

Table 1: Add the caption information: "Number of patients and respective accuracy (%)"

Answer: Added (line 158).

To conect the "previous" results with Lipids analys the phrase "The diagnosis of cervix lesions based on its lipid composition was considered in the present study" - should be rewritten

Answer: The phrase was replaced on “Diagnostic potential of cervix lipid composition was studied.” (line 172-173).

Line 108: Suggestion: this description can be improved, Avoid introduce to many values in text in sequence - It is already shown in the table, and comment the general behaviour - keep the table as reference

Answer: Description of p-values were removed from the text (line 184)

The caption of Figure 1 and 2 could be improved by describing TG, SM, and so on....

Answer: Descriptions of TG, SM, etc. were added in the caption of Figure 1 and 2  (lines 212-216 and lines 219-223).

Table 2 - please provide the values in %

Answer: Sensitivity and specificity in Table 2 were recalculated in %.

2.2. I think you can rewrite, that paragraph is not necessary as it is written.

Answer: Paragraph 2.2 was expanded.

Most of the readers probably don´t know R or want to use so Could you provide the code as support info? Or did you use a specific library? can you make a more detailed description about that? - It is also important to people understand the data processing and verify/reproduce the study/application.

Answer: the R code was provided in pdf file (Supplementary file 1.).

Finally, I think you should reproduce this study by using Raman or FTIR spectroscopy to analyse the samples and built a prediction model with PCA+ML.

Answer: Thank you for this suggestion. We will try to perform this study on our future work.